# Modeling the Uncertainty with Maximum Discrepant Students for Semi-supervised 2D Pose Estimation

## Abstract

Semi-supervised pose estimation is a practically challenging task for computer vision. Although numerous excellent semi-supervised classification methods have emerged, these methods typically use confidence to evaluate the quality of pseudo-labels, which is difficult to achieve in pose estimation tasks. For example, in pose estimation, confidence represents only the possibility that a position of the heatmap is a keypoint, not the quality of that prediction. In this paper, we propose a simple yet efficient framework to estimate the quality of pseudo-labels in semi-supervised pose estimation tasks from the perspective of modeling the uncertainty of the pseudo-labels. Concretely, under the dual mean-teacher framework, we construct the two maximum discrepant students (MDSs) to effectively push two teachers to generate different decision boundaries for the same sample. Moreover, we create multiple uncertainties to assess the quality of the pseudo-labels. Experimental results demonstrate that our method improves the performance of semi-supervised pose estimation on three datasets. The project code and our dataset are publicly available on `https://github.com/Qi2019KB/MDSs/tree/master`.

## 1 Introduction

As an essential task in computer vision, pose estimation has been widely used in various fields such as action recognition, security monitoring, animal experimentation. Most pose estimation methods use MSE loss to transform keypoint location predictions into Gaussian heatmap regression. State-of-the-art methods (Zhang et al. (2020a), Zhang et al. (2020b), Huang et al. (2019), Li et al. (2019), Cheng et al. (2019), Chen et al. (2018), Xiao et al. (2018), Tang et al. (2018), Newell et al. (2017), Cao et al. (2017), Newell et al. (2016), Wei et al. (2016)) require a large amount of labeled data. However, labeling pose keypoints is a costly and time-intensive labor for many practical tasks. How to effectively train a model without sufficient labeled data has attracted attention in the field of pose estimation.

A simple idea is to utilize semi-supervised classification methods for the pose estimation task. Numerous outstanding research have emerged in the field of semi-supervised classification, which can be categorized into methods based on consistency constraints and pseudo-labels. For example, Miyato et al. (2018), Sajjadi et al. (2016) mainly employ rational stochastic transformations and perturbations to improve the gain, which are derived from consistency constraints. Meanwhile, Tarvainen & Valpola (2017), Laine & Aila (2016) utilize high-quality pseudo-label filters from the predictions of unlabeled samples. In addition, the two methods are combined for better performance, such as Zhang et al. (2021), Chen et al. (2020), Berthelot et al. (2019).

However, semi-supervised classification methods commonly evaluate the quality of generated pseudo-labels based on confidence, such as Ke et al. (2019), which cannot be applied to semi-supervised pose estimation. In the pose estimation task, confidence can only be used to obtain the keypoint locations of the heatmap predicted by the model, not to evaluate the quality of the prediction. As shown in Fig. 1, with the increase of epochs, the model's prediction confidence of this key point gradually increases, but the error of this prediction does not effectively decrease. This phenomenon indicates that confidence in pose estimation is not directly related to the quality of pre-

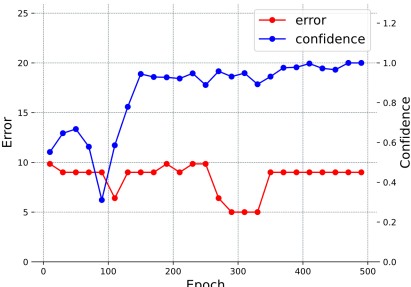 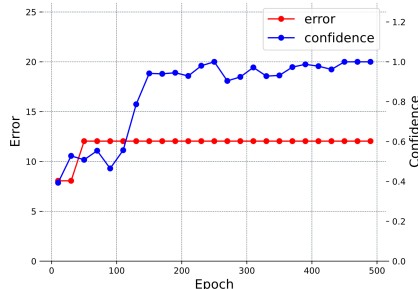

Figure 1: Error and confidence curves for two keypoint pseudo-labels, 1106_5 and 2077_4. It suggests that confidence is not directly related to the accuracy of the prediction itself. Therefore, confidence does not effectively assess the quality of pseudo-labels.

dictions from the model, and thus cannot be used to assess the quality of pseudo-labels. With these noisy and even erroneous pseudo-labels, the performance naturally deteriorates. That is, the performance of semi-supervised learning is lower than that of supervised learning with the same amount of labeled data. Therefore, effectively selecting high-quality pseudo-labels in semi-supervised pose estimation is an essential and challenging task.

Therefore, we construct a dual mean-teacher structure to evaluate the quality of the pseudo-labels by computing the distance between the predicted keypoint positions of multiple pose estimators to measure the discriminative power of the model on the samples. Furthermore, high-quality pseudo-labels are added to the training dataset to gradually train the model. To avoid coupling between multiple pose estimators, we introduce a simple and effective method to build the maximum difference regrestors (MDRs) by adding a parameter adversarial mechanism between two students. When the parameters of each regressor are different, it is possible to ensure that their decision boundaries do not overlap, thus helping to ensure the accuracy of the evaluated uncertainty. Meanwhile, the parametric adversarial mechanism can balance the performance of the two MDSs to guarantee the quality of pseudo-labels. The key contributions of this work are:

- We propose a semi-supervised pose estimation method based on the Dual Mean-Teacher framework. We construct multiple uncertainties to evaluate the quality of pseudo-labels generated by the model and efficiently extract high-quality pseudo-labels for further model training.

- We propose a model-parameter adversarial mechanism to effectively increase diversity and avoid the degradation of uncertainty assessment quality caused by the coupling of multiple pose regressors.

- Our experiments show that MDSs can effectively select high-quality pseudo-labels without using confidence as an evaluation criterion and achieve better performance on the three datasets.

## 2 RELATIVE WORK

### 2.1 2D POSE ESTIMATION

In terms of organization of supervised information, 2D pose estimation methods can be divided into keypoint regression-based methods and heatmap-based methods. For example, Toshev & Szegedy (2014) directly regress the keypoint positions using the coordinate values of the keypoints as supervision information. Moreover, Huang et al. (2019), Newell et al. (2016) and Wei et al. (2016) are heatmap based keypoint estimation methods. The keypoint coordinate values are converted into heatmap as supervised information to train the model, and then the appropriate position is selected from the heat output by the model as the position prediction of the keypoints according to confidence. We choose heatmap-based methods because direct coordinate regression is more difficult and less favorable for semi-supervised pose estimation.

## 2.2 SEMI-SUPERVISED LEARNING

Many excellent semi-supervised methods have emerged in the field of classification. For example, Miyato et al. (2018), Sajjadi et al. (2016) are consistency constraint based methods. Crucial to this approach is how to efficiently improve the effect of consistency constraints through reasonable data augmentation and perturbation, thus obtaining more supervised information from unlabeled data to improve model performance. While Tarvainen & Valpola (2017), Laine & Aila (2016), based on the idea of self-training, generate pseudo-labels from the model to supervise their training. So this approach relies on how to generate and select high-quality pseudo-labels. In addition, the two methods are combined for better performance, such as Zhang et al. (2021), Chen et al. (2020), Berthelot et al. (2019). The core idea of Ge et al. (2020) is the composition of two mean-teachers providing each other with pseudo-labels. Our work differs from it in that we maximize the difference in parameters between the two students to improve the discrepancy between the two models. We combine the two using the Mean-Teacher. Our model not only obtains supervised information based on consistency constraints from unlabeled data, but also generates high-quality pseudo-labels through mutual generalization of teacher and student models.

## 2.3 QUALITY ASSESSMENT OF PSEUDO-LABELS

Semi-supervised classification methods normally filter self-supervised information by combining uncertainty and confidence, such as Ke et al. (2019), Sohn et al. (2020). This is because uncertainty is commonly quantified by evaluating the consistency of multiple predictions for the same sample, which requires the results of each prediction to be plausible. However, unlike the classification task, confidence in pose estimation can only be used to obtain keypoint locations from heatmaps, not to evaluate the quality of model predictions.

Yang et al. (2021) calculates the uncertainty of the pseudo-labels in a predictive way. In contrast, our work proposes that confidence should no longer be used as an evaluation standard when evaluating the quality of pseudo-labels, and we use the tri-uncertainty to replace confidence.

Therefore, our method evaluates the quality of pseudo-labels through uncertainty without confidence. To ensure uncertainty accuracy, we construct an uncertainty evaluation network based on the dual Mean-Teacher structure and add a model parameter adversarial mechanism so that the decision surfaces between multiple pose estimators do not overlap.

## 3 PROPOSED METHOD

### 3.1 SEMI-SUPERVISED POSE ESTIMATION

The supervised pose estimation is to achieve accurate predictions of the position of $K$ keypoints. Most state-of-the-art methods convert the position of keypoints as heatmap $\mathbf{H}$. $\mathbf{H}$ is a distribution, encoding the possibility that each pixel in image $\mathbf{I}$ is the keypoint. Denote the labeled training dataset as $\mathcal{L} = \{(\mathbf{I}^l, \mathbf{H}^l)\}_{l=1}^{N^L}$. As a result, the prediction of keypoints' locations is converted as the Gaussian heatmap regression by minimize the MSE loss between the prediction and its ground-truth. The **pose loss for labeled data** is as follows:

$$L_p^l = \mathbb{E}_{\mathbf{I} \in \mathcal{L}} \|\mathbf{f}(\mathbf{I}|\theta) - \mathbf{H}\|^2, \tag{1}$$

where $\mathbf{f}(\cdot|\theta)$ is the model.

Comparing with supervised learning, the semi-supervised pose estimation solves the problem of less amount of labeled data. Denote the unlabeled training dataset as $\mathcal{U} = \{\mathbf{I}^u\}_{u=1}^{N^U}$. For unlabeled data, the ground-truth is the pseudo-label $\hat{\mathbf{H}}$ generated in the semi-supervised learning. The **pose loss for the unlabeled data** is as follows:

$$L_p^u = \mathbb{E}_{\mathbf{I} \in \mathcal{U}} \|\mathbf{f}(\mathbf{I}|\theta) - \hat{\mathbf{H}}\|^2, \tag{2}$$

We use Mean Teacher (MT) Tarvainen & Valpola (2017) in semi-supervised pose estimation. In MT, the teacher generates the pseudo-labels for unlabeled data to train the student, the parameters of

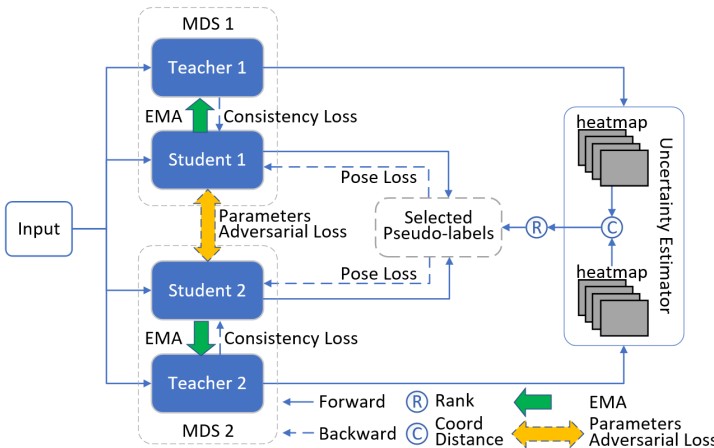

Figure 2: Overview of the proposed maximum discrepant students (MDSs) framework.

teacher are obtained from student by the Exponential Moving Average (EMA) strategy. The **EMA** is as follows:

$$\theta_T^t = \alpha\theta_T^{t-1} + (1 - \alpha)\theta_S^t, \tag{3}$$

where $\theta_T^t$, $\theta_S^t$ are the teacher's and student's parameters in $t$ epoch, respectively. $\alpha$ is a smoothing coefficient hyper-parameter. Taking advantage of EMA, teacher can generate more accurate pseudo-labels.

The quality of the pseudo labels is related to the predictive power of the generator at each epoch. However, models in semi-supervised learning typically use a single regressor to generate pseudo-labels, such as the teacher in MT. A single regressor obtains barely additional information. With a small amount of labeled data, the model tends to under- or over-fit, and thus generate noisy pseudo-labels. Therefore, an effective quality evaluation method for pseudo-labels is needed to pick up high-quality pseudo-labels to train the model.

### 3.2 Maximum Discrepant Students

The MDSs framework is illustrated in Fig. 2. It is composed of two MDS modules, and an uncertainty estimator $E$. Both students and teachers use the same pose estimation model. In each MDS, the teacher predicts the pseudo-label to train the student and uses the Exponential Moving Average (EMA) weights of the student. The uncertainty estimator $E$ evaluates the uncertainty of the pseudo-label by calculating the distance between two predictions from teachers $T_1$ and $T_2$. We maximize the discrepancy of two students $S_1$ and $S_2$ by the **parameters adversarial loss**, as equation 6.

Each epoch, two teachers generate the pseudo-labels of unlabeled data. The uncertainty estimator $E$ estimates the uncertainty of these pseudo-labels, and evaluates the quality of these pseudo-labels by the uncertainty. The top k high-quality pseudo-labels are added into labeled dataset. Then two students are trained by this expanded labeled dataset.

### 3.3 Objective Function

In supervised pose estimation, the MSE loss is used to train the model. In MDSs, we directly use the ground-truth to calculate pose loss for labeled data. For unlabeled data, we use the pseudo-labels generated by teacher $T_2$ to calculate the **pose loss** of student $S_1$, which is as follows:

$$L_{p_{S_1}} = \mathbb{E}_{\mathbf{I}\in\mathcal{L}}\|\mathbf{f}(\mathbf{I}|\theta_{S_1}) - \mathbf{H}\|^2 + \mathbb{E}_{\mathbf{I}\in\mathcal{U}}\|\mathbf{f}(\mathbf{I}|\theta_{S_1}) - \hat{\mathbf{H}}_{T_2}\|^2, \tag{4}$$

where $\theta_{S_1}$, $\theta_{T_2}$ are the parameter of the student $S_1$ and the teacher $T_2$, respectively. $\hat{\mathbf{H}}_{T_2}$ is the pseudo-label generated by the teacher $T_2$.

Correspondingly, the **pose loss** of student $S_2$ is calculated with the pseudo labels generated by teacher $T_1$, which is as follows

$$L_{p_{S_2}} = \mathbb{E}_{\mathbf{I} \in \mathcal{L}} \| \mathbf{f}(\mathbf{I}|\theta_{S_2}) - \mathbf{H} \|^2 + \mathbb{E}_{\mathbf{I} \in \mathcal{U}} \| \mathbf{f}(\mathbf{I}|\theta_{S_2}) - \hat{\mathbf{H}}_{T_1} \|^2, \tag{5}$$

where $\theta_{S_2}$, $\theta_{T_1}$ are the parameter of the student $S_2$ and the teacher $T_1$, respectively. $\hat{\mathbf{H}}_{T_1}$ is the pseudo-label generated by the teacher $T_1$.

In real life, when two experts in the same field have different answers to the same question, it is most likely that the question itself is very uncertain. Inspired by this, the MDSs framework is based on two constraints.

On one hand, two teachers are used to evaluate the uncertainty of the same sample. To achieve this goal, the first constraint is that two teachers should have balanced performance, as equation 4 and equation 5.

On the other hand, two students should keep divergences. So, another constraint is the *parameter adversarial loss*, through which we ensure that the two experts are not homogenized. To maximize the discrepancy between two MDSs, we minimize the cosine distance of students $S_1$ and $S_2$ by the **parameters adversarial loss**:

$$L_a = \frac{\theta_{S_1} \cdot \theta_{S_2}}{\|\theta_{S_1}\| \|\theta_{S_2}\|}, \tag{6}$$

Otherwise, following the semi-supervised approach, the **consistency loss** to updating $S_1$ and $S_2$ is as follows:

$$L_c = \mathbb{E}_{\mathbf{I} \in \mathcal{L} \cup \mathcal{U}} \| f(\mathbf{I}|\theta_{S_1}) - f(\mathbf{I}|\theta_{T_1}) \|^2 + \mathbb{E}_{\mathbf{I} \in \mathcal{L} \cup \mathcal{U}} \| f(\mathbf{I}|\theta_{S_2}) - f(\mathbf{I}|\theta_{T_2}) \|^2, \tag{7}$$

We summarize the objective function as follows:

$$L = \lambda_p (L_{p_{S_1}} + L_{p_{S_2}}) + \lambda_a L_a + \lambda_c L_c, \tag{8}$$

where $\lambda_p$, $\lambda_a$ and $\lambda_c$ are the weights to balance all losses.

### 3.4 HIGH-QUALITY PSEUDO-LABELS

Crucial in the semi-supervised pose estimation task is how to judge the quality of a pseudo-label.

First, we take the prediction $f(\mathbf{I}|\theta_T)$ of the teacher $\theta_T$ on the raw sample $\mathbf{I}$ as the pseudo-truth value of the pseudo-label. This is because, in pose estimation tasks, the prediction accuracy of raw samples is much higher than that of augmented samples, which are generated by data augmentation or perturbation.

Second, to evaluate the quality of pseudo-labels, we consider the following two points:

- For high-quality pseudo-labels, the model should have predictive consistency for different augmented samples of the raw sample. A model has a high epistemic capacity on raw samples if its prediction uncertainty on augmented samples is low. In turn, high uncertainty implies low epistemic capacity of the model for the raw sample. Therefore, we obtain $M$ randomly augmented samples of this raw sample and calculate the distance between the predictions of these augmented samples to quantify the uncertainty of the corresponding pseudo-label. The lower the uncertainty, the higher the quality of the pseudo-labels.
- For high-quality pseudo-labels, the model should have predictive stability for the raw sample and its augmented samples. Therefore, we evaluate the quality of the pseudo-labels by computing their uncertainties via a moving weighted average.

### 3.5 THE QUANTIFICATION OF UNCERTAINTY

To better assess the quality of pseudo-labels, we construct triplet uncertainties, which are augmented internal uncertainty $unc_{aug}^{int}$, augmented external uncertainty $unc_{aug}^{ext}$, and raw internal uncertainty $unc^{ext}$.

The augmented internal uncertainty is the average distance between the predictions $\{\widetilde{\mathbf{p}}\}_{m=1}^{M} = \{f(\widetilde{\mathbf{I}}^m|\theta_T)\}_{m=1}^{M}$ generated by the teacher on the $M$ augmented samples.

$$\text{unc}_{\text{aug}}^{\text{int}} = \text{M}(\text{D}(\text{C}(\{\widetilde{\mathbf{p}}\}_{m=1}^{M}))), \tag{9}$$

where $\text{M}(\cdot)$ is to calculate the mean value from a distances set, D is a function to calculate the Euclidean distance between two coordinates, C is a combinatorial function, selecting 2 prediction of all, non-repeatedly.

The augmented external uncertainty is the distance between the mean prediction $\bar{\widetilde{\mathbf{p}}}_{\theta_{T_1}}$ of one teacher $T_1$ and the mean prediction $\bar{\widetilde{\mathbf{p}}}_{\theta_{T_2}}$ of another teacher $T_2$ on the same set of augmented samples.

$$\text{unc}_{\text{aug}}^{\text{ext}} = \text{D}(\bar{\widetilde{\mathbf{p}}}_{\theta_{T_1}}, \bar{\widetilde{\mathbf{p}}}_{\theta_{T_2}}), \tag{10}$$

The raw external uncertainty is the distance between the prediction $\mathbf{p}_{\theta_{T_1}}$ of one teacher and the prediction $\mathbf{p}_{\theta_{T_2}}$ of another teacher on the same raw sample.

$$\text{unc}^{\text{ext}} = \text{D}(\mathbf{p}_{\theta_{T_1}}, \mathbf{p}_{\theta_{T_2}}), \tag{11}$$

When selecting high-quality pseudo-labels, we require that all three uncertainty values should be smaller than the threshold $\epsilon = 3.0$.

## 4 EXPERIMENTAL RESULTS

### 4.1 DATASET

The **FLIC** Mathis et al. (2018) dataset consists of 5003 images taken from movies, including 3987 training data and 1016 test data. Most images have a single instance and a few have multiple instances. Each instance contains 11 keypoints. In our experiments, we use single instance samples, of which 1000 are used as the training set and 500 are used as the testing set. Fifty percent of the samples in the training set are labeled. We selected keypoints on the left shoulder and left waist to participate in the experiment. When computing the PCK, we use the distance between the left shoulder and the left waist as the scale factor. The metric of PCK@0.2 is reported.

The **Pranav** Sapp & Taskar (2013) dataset is a public mouse dataset consisting of 1000 images with one instance per image. Each mouse instance has four keypoints such as nose, left ear, right ear, and tail root. In our experiments, we use 100 samples as the training set and 500 samples as the test set. Thirty percent of the samples in the training set are labeled. When computing the PCK, we use the distance between the left ear and the right ear as the scale factor. The metric of PCK@0.2 is reported.

The **Mouse**[1] dataset was captured by ourselves from real experimental environments. The dataset contains 1000 RGB images with 256 x 256 pixels. The pose annotation of the image consists of nine keypoints such as nose, left eye, right eye, left ear, right ear, head, tail root, tail and tip. In our experiments, we use 100 samples as the training set and 500 samples as the test set. Thirty percent of the samples in the training set are labeled. When computing the PCK, we use the distance between the left ear and the right ear as the scale factor. The metric of PCK@0.2 is reported.

### 4.2 IMPLEMENTATION DETAILS

We experimented with the Stacked Hourglass (HG), Mean Teacher (MT), Easy-Hard Augmentation (EHA) and our MDSs on the FLIC, Pranav and Mouse datasets, respectively. Among the methods, HG is supervised learning and the remaining three methods are semi-supervised learning.

**Stacked Hourglass (HG). Newell et al. (2016)** In the HG, the number of stacks is 3, and other settings are the same as these in Newell et al. (2016).

---

[1] This dataset is available at this project address.

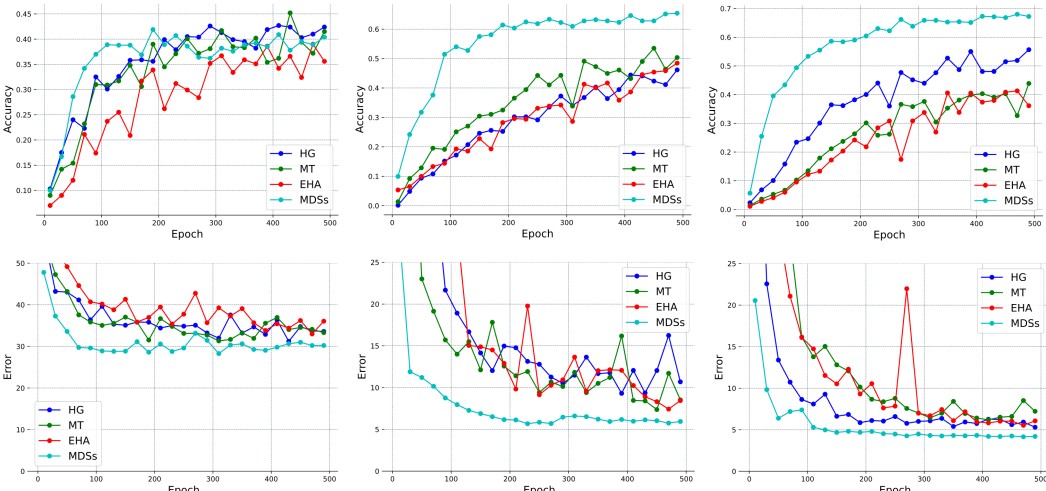

Figure 3: The error and PCK@0.2 of different methods on three datasets, respectively. HG Newell et al. (2016) is supervised learning. MT Tarvainen & Valpola (2017), Easy-Hard Augmentation (EHA) Xie et al. (2021) and MDSs are semi-supervised learning. **Left Colomn:** Result on FLIC dataset, 50% of the 1000 samples are labeled. **Middle Colomn:** Result on Pranav dataset, 30% of the 100 samples are labeled. **Right Colomn:** Result on Mouse dataset, 30% of the 100 samples are labeled.

**Mean Teacher (MT). Tarvainen & Valpola (2017)** In the MT, we change the classification model to Stacked Hourglass Newell et al. (2016). Moreover, the number of stacks is 3. The initial weight of the consistency loss between teacher and student is 0.0, and after 50 epochs, it increases from 0.0 to 20.0. The initial value of pose loss is the constant 10.

**Easy-Hard Augmentation (EHA). Xie et al. (2021)** EHA is based on the idea of using the predictions of easy-to-augment samples to supervise the predictions of hard-to-augment samples. Thus, as mentioned Xie et al. (2021), we constructed easy-hard augmentation pair with random rotations of 30ř and 60ř. The initial weight of the consistency loss between Easy and Hard models is 0.0, which is increased from 0.0 to 20.0 after 50 epochs. The weight of pose loss is a constant 10.

**MDSs.** We use Stacked Hourglass as the pose estimation model, the number of stacks is 3. In the pseudo-label generation process, we generate 5 different data-augmented samples for each sample to calculate its uncertainty. Moreover, when selecting high-quality pseudo-labels, the triple-uncertainty's common threshold $\epsilon$ is 3.0. The value of $\lambda_p$ is the constant 10. The initial value of $\lambda_a$ is 0.005 and after 500 epochs decreases to 0.0. The initial value of $\lambda_c$ is 0.0, and after 50 epochs, it increases from 0.0 to 20.0.

In all of the above experiments, all input images are resized to $256 \times 256$ pixels. Moreover, we use the data augmentation that includes random rotation $(+/-30$ degrees) (except EHA, which has been indicated in its paragraph), and random scaling $(0.75-1.25)$ and random horizontal flip. The training batch size is two and the total number of training epochs is 500.

## 4.3 COMPARISONS WITH OTHER METHODS

As shown in Table 1, MDSs achieve the best results, both in terms of error and PCK@0.2, on both Mouse and Pranav datasets. MDSs achieve better results on the FLIC dataset in terms of error, with at least a 3-points improvement over the other methods. The error and PCK@0.2 curves for different models on three datasets shown in Fig. 3. It can be seen that, on Pranav and Mouse datasets, MDSs perform considerably better than the other methods in terms of both PCK@0.2 and Error curves throughout the training period. On the FLIC dataset, the lack of labeled data makes it difficult to generate pseudo-labels with high accuracy, as the model's knowledge of the samples is not sufficient. However, even then, the error curve of the MDSs is still better than others.

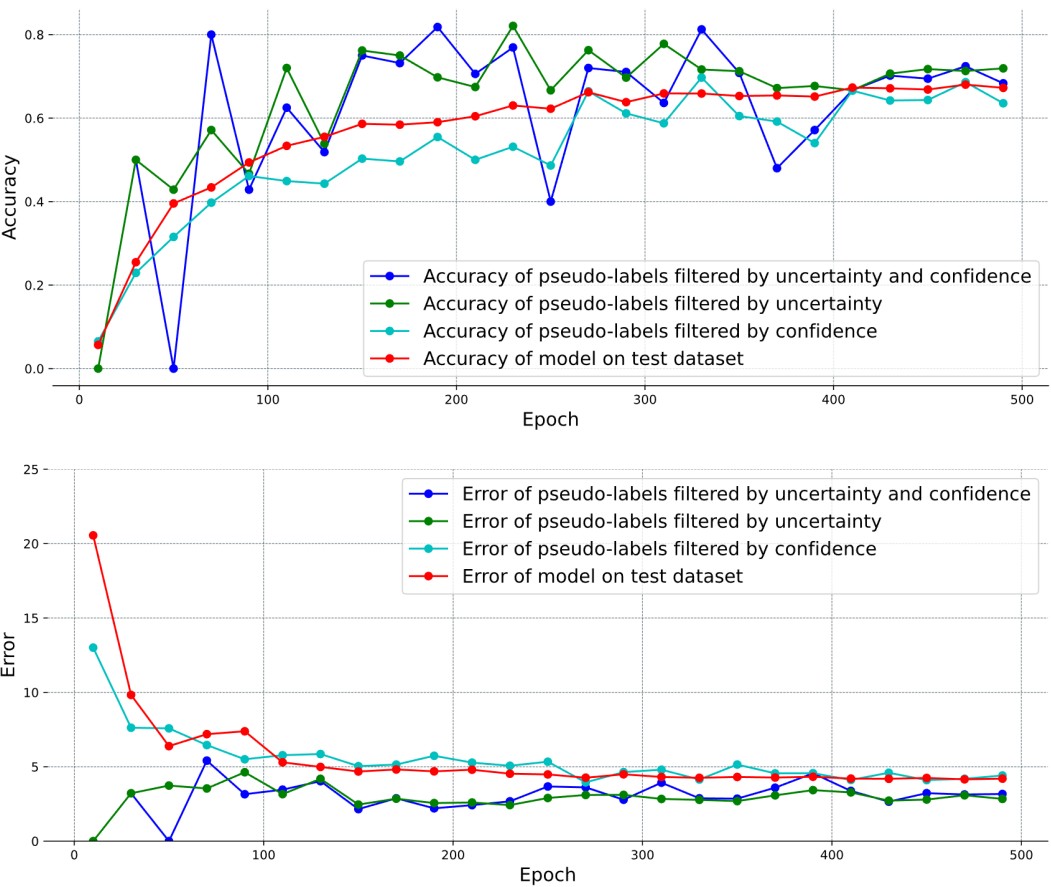

Figure 4: The PCK@0.2 (**Top**) and error (**Bottom**) of different methods on the Mouse datasets.

Table 1: MSE and PCK@0.2 of different methods on FLIC, Pranav, Mouse Dataset. "1000*0.5" represents the represents 1000 samples as the training set, 50% of which are labeled.

| Method | Training set | MSE | PCK@0.2 |
|---|---|---|---|
| supervised Newell et al. (2016) | | 29.70 | **0.477** |
| Mean-Teacher Tarvainen & Valpola (2017) | FLIC | 33.68 | 0.469 |
| EHA Xie et al. (2021) | 1000*0.5 | 32.67 | 0.428 |
| Ours | | **27.47** | 0.420 |
| supervised | | 9.39 | 0.485 |
| Mean-Teacher | Pranav | 6.54 | 0.554 |
| EHA | 100*0.3 | 7.18 | 0.502 |
| Ours | | **5.66** | **0.660** |
| supervised | | 5.29 | 0.583 |
| Mean-Teacher | Mouse | 4.75 | 0.627 |
| EHA | 100*0.3 | 5.18 | 0.482 |
| Ours | | **4.10** | **0.687** |

## 4.4 THE EFFECT OF UNCERTAINTY

To further illustrate the effectiveness of our method, we select a small amount of labeled data as unlabeled data to generate and select high-quality pseudo-labels. Moreover, calculating the error and PCK@0.2 of these pseudo-labels to observe the effect of uncertainty on the quality evaluation of pseudo labels. We performed the following experiments:

- In MDSs, triplet uncertainty and confidence are used to select high-quality pseudo-labels.
- In MDSs, triplet uncertainties are used to select high-quality pseudo-labels.
- In MDSs, confidence is used to select high-quality pseudo-labels.

The experimental results are shown in Figs. 4. On the one hand, the pseudo-label quality of the two methods using triplet uncertainty is much higher than the prediction performance of the current model on the test set. These results suggest that triplet uncertainty can effectively select high-quality pseudo-labels. On the other hand, the pseudo-label quality is similar for both methods using triplet uncertainty, while the method using confidence alone has the worst pseudo-label quality.

## 4.5 ABLATION STUDY

Table 2: Evaluation result of MDSs under different settings. PAL is the *parameter adversarial loss* of MDSs, as equation 6.

| Index | PAL | PAL Weight | PCK@0.2 |
|-------|-----|------------|---------|
| 1 | | | 0.675 |
| 2 | ✓ | 0.000005 | 0.687 |
| 3 | ✓ | 0.005 | **0.693** |

In order to illustrate the role of model parameter adversarial mechanism in MDSs, we ran experiments with different parameter adversarial loss weights (PAL Weight), and the experimental results are shown in Table 2. Use the parameter adversarial mechanism, the error and PCK@0.2 of the model are better than those without parameter adversarial mechanism.

## 5 CONCLUSION

In this paper, a Maximum Difference Student (MDSs) framework is proposed to effectively improve the performance of semi-supervised pose estimation tasks. In, evaluation metrics based on triplet uncertainty can effectively select high-quality pseudo-labels and completely replace confidence. The model parameter adversarial mechanism can further improve the consistency and stability of the predictions of the two teachers and effectively boost the performance of the model. Our method achieves better results on three datasets: FLIC, Pranav, and Mouse.

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
