# OpenReview forum: "Modeling the Uncertainty with Maximum Discrepant Students for Semi-supervised 2D Pose Estimation"
_ICLR.cc/2023/Conference — Submitted to ICLR 2023_

### Official Review · Reviewer_2eQb · 2022-10-18

**Confidence:** 4
**Clarity, Quality, Novelty And Reproducibility:** See above Strength And Weaknesses
**Correctness:** 3
**Technical Novelty And Significance:** 2
**Empirical Novelty And Significance:** 3
**Recommendation:** 5

**Strength And Weaknesses:**


Strengths:
1. Well written and easy to read.
2. Theperformance seem to be good.
Weakness:
1. Key references are missing. Using two student and teacher models to supervised each other have been explored in previous works.
Specifically, the motivation of [1] is to utilize the discrepancy between two models to generate soft pseudo labels, [2] models the uncertainty of the pseudo labels in a predictive way.
[1] Mutual Mean-Teaching: Pseudo Label Refinery for Unsupervised Domain Adaptation on Person Re-identification
[2] Uncertainty guided collaborative training for weakly supervised and unsupervised temporal action localization
2. Technique contributions are not clear. This paper seems to be the use of existing  techniques on a new task, although this is reasonable, it lacks task specific technique contributions.
3. Experimental evaluations are not convincing. (1) For semi supervise learning, a key metrich is the performance with different labeled data ratios, but this is missing in this paper.  (2) There are many ways for uncertainty estimation, the advantage of the way in this paper is not clear.
4. There are some typos in this paper, such are "the prediction of the hard-agumented sample [Paper]".

**Summary Of The Paper:**

This paper addresses the semi supervised pose estimation problem with the mean-teacher framework.
Specifically, the key insight is to maximum the discrepancy between two student models and use the corresponging teacher models to generate pseudo labels  to supervise each other.
Besides, a discrepancy loss is used to maximum the discrepancy between the weights of two student models, and an uncertainty estimation method is used to evaluate the uncertainty of pseudo labels

**Summary Of The Review:**

My main concern is the technique contribution and experiment evaluation.

---

> ### Author Response · Authors · 2022-11-19
> **Thanks for your feedback!**
>
> Thank you very much for your review. It means a lot to us.
>
> $ \cdot "Key \  references \  are \  missing \  ..." $
>
> -- Thanks for pointing out these additional references! We have updated our draft with a discussion surrounding this line of work.
>
> $ \cdot "Technique \  contributions \  are \  not \  clear  \  ..." $
>
> -- Thanks for pointing this out. We focus on how to better select high-quality pseudo-labels. Therefore, triplet uncertainty is proposed as a replacement for confidence to evaluate the quality of pseudo-labels.​
>
> $ \cdot "Experimental \  evaluations \  are \  not \  convincing \  ..." $
>
> -- Your opinion is pertinent. We do need to add more experiments to further refine our approach. However, due to the long duration of the experiment, it was not possible to complete the new experiment during the rebuttal phase. We will finished these experiments as soon as possible and add these experiments to the next version.
>
> $ \cdot "There \  are \  some \  typos \  in \  this \  paper \  ..." $
>
> -- Thank you very much for the problems you pointed out. We have rechecked the paper and revised some similar problems.

---

### Official Review · Reviewer_nhdF · 2022-10-19

**Confidence:** 3
**Correctness:** 2
**Technical Novelty And Significance:** 3
**Empirical Novelty And Significance:** 3
**Recommendation:** 3

**Clarity, Quality, Novelty And Reproducibility:**

- p.3, chapter 3.1:  How is y used - its in {0,1}^K, but should it not represent a keypoint location? Why does for x^u_j j runs from N+1 to N+M, when x^u and x^l are already differentiated with the superscript?  Further confusing, why do later in Eq. (4), x^l_i and x^u_i exist at the same time (i.e. labeled and unlabeled x with the same index)?
 x^l_i, y^l_i, and K are defined twice (once in the first, once in the second paragraph of Sec. 3.1).

- Eq.(8): L_p (the pose loss) is undefined (likely L_p_S1 + L_p_S2?)

- Eq.(9): I am unsure how the function C works. Does it select only two predictions, all (distinct) pairs of predictions, does it sample a set of pairs (and if yes, how many)?
 Also, Eq. (1),(4),(5) indicate that the output of f(x|theta) is a heatmap. Yet in Eq.(9), it appears to be used as producing a coordinate.
 x ~ ^m is undefined. What is the difference between {p ~ }_m=1^M in Eq.(9) and p~_\theta_T_1 in Eq.(10)? The notation appears to suddenly switch from f(x|theta) to p_\theta.

- p.5: The parameter adversarial loss (Eq. (6)) does not really enforce non-homogenous students, since there is no semantic 1-to-1-correspondence between their parameters. Two students can, for example, contain permutations of the same feature maps that have a small L_a, even though they compute the same function.  While the ablation in Table 2 shows a positive impact of this loss, I am unsure about its interpretation and motivation.

    Further, the evaluation in Table 2 falls short of optimizing the PAL weight. Larger weights than the presented optimum of 0.005 could lead to even better results.

    If the "PAL weight" is "\lambda_a", I'd recommend to re-use that symbol in Table 2.

- The method as presented is likely not reproducible. Details on the network architecture and training strategy and parameters are missing. Additionally, several hyperparameters are not given. Eq.(8): The values of \lambda_* are not given (except for \lambda_a in Table 2?). The threshold \epsilon from Sec. 3.5 is not given, neither is a way of deriving it (it probably depends on the dataset, since the unc-Values from Eq.(9),(10),(11) are not normalized).

- There appear to be no citiations for the datasets (FLIC, Pranav, Mouse).

- The compared-with methods appear rather "old" for deep-learning based methods (2016, 2017 and one from 2021). It is unclear why the results of other more recent work is not listed in the results of Table 1.

- While the experiments show an improvement over the baseline "Mean-Teacher", it is not demonstrated that this improvement is due to the modified training architecture, or simply because the proposed network has twice the capacity compared to Mean-Teacher.



Minor issues (typos etc.). No need to respond.

- Multiple references are made to "Eq. equation XX" - "Eq." and "equation" are redundant and one of them should be deleted.

- p.3, "is the corresponding *the* ground-truth" - delete second the

- p.4, invalid reference: "T_1 and T_2, as Eq. equation ??"

- p.7, missing \cite: "of the hard-augmented sample [Paper]"


**Strength And Weaknesses:**


**Novelty**: The idea of using an ensemble of self-supervised / semi-supervised networks is well thought of and an interesting direction. The method itself could generalize to other settings as well.

**Presentation**: The manuscript is in parts unclear regarding notation (see below). Some parts of the manuscript appear unfinished (invalid references, see below).

**Reproducibility**: I believe it is unlikely that the work can be reproduced. Details on the network architecture, training setup and hyperparameters are missing (see below).

**Experimental Setup**: The method is evaluated on three smaller datasets (which are not cited) and compared with mostly older methods. It remains unclear how it would perform on larger datasets against state of the art. Additionally, the experiments do not show if the improvement over the baseline method is due to the proposed architectural improvements or simply due to the doubeling of the network's capacity.

**Results**: The results are good and show a significant improvement over the compared-with methods.


**Summary Of The Paper:**

This manuscript recommends an ensemble-like approach for semi-supervised pose estimation. A teacher-student setting where self-supervised pseudo-labels are created by the teacher for unlabeled data is extended to use two teacher-student pairs. The discrepancy between the two teachers is used to assess the quality of the pseudo-labels and to select pseudo-labels that are more likely to be correct. The discrepancy between the students is used in a loss function. The method is evaluated on three datasets, showing improved results.

**Summary Of The Review:**

While the manuscript explores an interesting, general idea, it lacks important details and clarity and does not sufficiently demonstrate that the architectural improvements lead to improved results.

---

> ### Author Response · Authors · 2022-11-19
> **Thanks for your feedback!**
>
> $ \cdot "p.3, \  chapter \  3.1: \  How \  is \  y \  used \  ..."$
>
> -- Thanks for pointing this out.We have updated the relevant formula in the draft. Please refer to Chapter 3 in the draft.
>
> $ \cdot "Eq.(8): \  L_p \  (the \  pose \  loss) \  ..."$
>
> -- Yes, we have updated the draft to make this more clear. Please refer to Chapter 3.3 in the draft.
>
> $ \cdot "Eq.(9): \   I \   am \   unsure \   how \   the \   function \   C \   works \  ..."$
>
> -- Thanks for pointing this out.We have updated the relevant formula in the draft. Please refer to Chapter 3 in the draft.
>
> $ \cdot "p.5: \  The \  parameter \  adversarial \  loss \  (Eq. \  (6)) ..."$
>
> -- Thank you very much for the problems you pointed out. In order to solve the overlapping problem of decision surfaces, the parameter adversarial loss (Eq. (6)) was considered. Two models with significantly different parameters are like two experts that achieve differentiated predictions for the same sample by generating different decision surfaces.
>
> $ \cdot "The \  method \  as \  presented \  is \  likely \  not \  reproducible \  ..."$
>
> -- Thanks for pointing this out. ​In the implementation section, we have added details about the setting of the hyperparameters and the training strategy of the project. We have released the project code in draft.
>
> $ \cdot "There \  appear \  to \  be \  no \  citiations \  for \  the \  datasets  ..."$
>
> -- Thanks for pointing out these related work. ​In the draft, we have added references to public datasets and download addresses for our own dataset.
>
> $ \cdot "The \  compared-with \  methods \  appear \  rather \  old \  ..."$
>
> -- Thanks for your helpful suggestions. As you mentioned, we will do more experiments with other more recent models. However, due to the long duration of the experiment, it was not possible to complete the new experiment during the rebuttal phase. We will add these experiments to the next version.
>
> $ \cdot "While \  the \  experiments \  show \  an \  improvement \  over \  the \  baseline \  Mean-Teacher \  ..."$
>
> -- Thank you very much for the problems you pointed out. However, due to the long duration of the experiment, it was not possible to complete the new experiment during the rebuttal phase. We will conduct relevant comparative experiments to further verify our method. These experiments will be added to the next version.

---

### Official Review · Reviewer_WbjR · 2022-10-24

**Confidence:** 4
**Correctness:** 3
**Technical Novelty And Significance:** 3
**Empirical Novelty And Significance:** 2
**Recommendation:** 5

**Clarity, Quality, Novelty And Reproducibility:**

The idea of using mean-teacher networks looks novel and intersting. The paper is including details for reproducing their results.

**Strength And Weaknesses:**

The proposed method outperforms the other methods in terms of MSE score and the stability i.e., much smaller fluctuations on the error curve and unlike the other methods, it does not use the confidence for evaluation of quality of pseudo-labels.

The authors use 3 datasets in which two of them are much smaller dataset (Mouse, Pranav) than the other one (FLIC). In small datasets in terms of PCK@0.2 their approach outperforms the other methods whereas in the large dataset it has the lowest PCK@0.2 among all which raises the question of generalizability of this method to larger datasets. Besides, the error curve on only one dataset (Mouse) is reported.


**Summary Of The Paper:**

This paper proposes a semi-supervised pose estimation method based on the Dual Mean-Teacher framework. It aims to efficiently select high quality pseudo-labels. The framework, as the name suggests, is decomposed of two Mean-Teacher modules where teacher generates pseudo-labels by taking the parameters from student with Exponential Moving Average strategy to train the student. MSE loss along with parameter adversarial loss to keep the divergence and consistency loss is used to train the model.


**Summary Of The Review:**

Even though the quantitative results are good; I am not very convinced with the experimental settings yet: The method is working well for small-scale data; while it is showing the inferior results for large-scale data. Also, measures are inconsistent from dataset to dataset. I’d like to request authors’ thought on these points. Currently, I’m marginally below the threshold for this paper.

---

> ### Author Response · Authors · 2022-11-19
> **Thanks for your feedback!**
>
> $ \cdot "I \  am \  not \  very \  convinced \  with \  the \  experimental \  settings \  yet: The \  method \  is \  working \  well \  for \  small-scale \  data, \  while \  it \  is \  showing \  the \  inferior \  results \ ..." $
>
> -- Thank you! Your opinion is pertinent. We do need to add more experiments to further refine our approach. However, due to the long duration of the experiment, it was not possible to complete the new experiment during the rebuttal phase. We will finished these experiments as soon as possible and add these experiments to the next version.
>
> $\ cdot "Also, \  measures \  are \  inconsistent \  from \  dataset \  to \  dataset \  ..."$
>
> -- Sorry for that we didn't explain that clear in the draft. For each different dataset, we use a different norm factor when calculating the PCK. As in Section 4.1, in the FLIC dataset, we use the distance between the left shoulder and the left waist as the norm factor. And, in Pranav datset, we use the distance between the left ear and the right ear as the norm factor. Therefore, the evaluation criteria are different for each dataset.

---

### Official Review · Reviewer_rtkR · 2022-10-24

**Confidence:** 2
**Correctness:** 3
**Technical Novelty And Significance:** 3
**Empirical Novelty And Significance:** 3
**Recommendation:** 5

**Clarity, Quality, Novelty And Reproducibility:**

The presented approach lacks implementation details for reproducibility. The clarity part of the paper needs further work, i.e. definitions and method explanation, as well as overall writing needs some polishing. Finally, the main novelty of the paper is to estimate the quality of the pseudo-labels using uncertainty measures instead of confidence scores. This is a valid point but it does not deliver a major novelty.

**Strength And Weaknesses:**

Strength:
- The paper studies an open problem, namely semi-supervised 2D pose estimations.
- The proposed method is simple, straightforward and easy to develop.
- The ablation studies are thorough.

Weaknesses:
- The results shown do not clearly show a significant improvement compared to other methods in most of the examples, e.g. Xie et al. Table1).
- The presented dual teacher framework, mean teacher, and the uncertainty estimation protocol are known. Thus, the paper lacks a major contribution.
- Evaluation on more human pose data, e.g. MPII or COCO, would  be helpful for comparing with a wider range of methods.
- The paper writing has space for improvement.

**Summary Of The Paper:**

This paper proposes a semi-supervised pose estimation approach based
the dual mean-teacher framework. To evaluate the quality of the generated pseudo-labels, the approach makes use of the teacher networks' predicted uncertainty, where the uncertainty aims to select high-quality pseudo-labels, to be added to the unlabeled data and gradually train the model. Next, the paper presents an adversarial mechanism to build the maximum discrepancy within the student networks. Finally, the proposed approach is evaluated on three datasets (FLIC, Pranav, and Mouse), where it shows solid performance.

**Summary Of The Review:**

The paper presents an interesting idea but the novelty is limited. Moreover, the evaluation would be more beneficial with a more decent comparison with prior works on at least an additional standard dataset. Finally, the paper writing needs further work.

---

> ### Author Response · Authors · 2022-11-19
> **Thanks for your review!**
>
> $ \cdot "The \  results \  shown \  do \  not \  clearly \  show \  a \  significant \  improvement ..."$
>
> -- Thanks for pointing this out.Compared to the remaining two datasets, FLIC is more difficult. When the labeled data is small, the epistemic power of the model over the sample is too low and thus it has a large impact on the uncertainty of the model-based prediction. In other semi-supervised methods, it is common to select at least 1000 labeled samples from the FLIC dataset, and 250 in ours. On the remaining two datasets, the advantage of our approach over the other methods is more evident, as the datasets are easier and thus the model is less affected by less labeled data. We do need new experiments using more labeled data on the FLIC dataset to compare our approach with other baselines. However, due to the long duration of the experiment, it was impossible to complete the new experiment during the rebuttal phase. We will complete these experiments as soon as possible and add them to the next version. Based on the above considerations, our method can reflect more advantages in the FLIC dataset when the amount of labeled data increases.
>
> $ \cdot "The \  presented \  dual \  teacher \  framework, \  mean \  teacher, \  and \  the \  uncertainty \  estimation \  protocol \  are \  known \  ..."$
>
> -- Thank you very much for the problems you pointed out. What you say makes sense. We focus on extracting high quality pseudo tags from semi-supervised pose estimation. Therefore, we found that in the semi-supervised learning, the confidence was not effective in evaluating posture pseudo-labels, driven by the small number of label data. Therefore, we propose a quality evaluation method based on uncertainty, which can better replace confidence.
>
> $ \cdot "Evaluation \  on \  more \  human \  pose \  data, \  e.g. \  MPII \  or \  COCO \  ... " $
>
> -- Your opinion is pertinent. We do need to add more experiments to further refine our approach. However, due to the long duration of the experiment, it was not possible to complete the new experiment during the rebuttal phase. We will finished these experiments as soon as possible and add these experiments to the next version.
>
> $ \cdot "The \  paper \  writing \  has \  space \  for \  improvement."$
>
> -- Thank you for pointing out this problem. We have corrected some errors in the latest version.
>
> $ \cdot "The \  presented \  approach \  lacks \  implementation \  details \  for \  reproducibility \  ..."$
>
> -- Thank you very much for the problems you pointed out. In the implementation section, we have added details about the setting of the hyperparameters and the training strategy of the project. We have released the project code in draft.

---

### Decision · Program_Chairs · 2023-01-20

**Decision:**

Reject

**Justification For Why Not Higher Score:**

This paper receives 3x marginally below the acceptance threshold and 1x reject, not good enough.

**Justification For Why Not Lower Score:**

NA

**Metareview: Summary, Strengths And Weaknesses:**

This paper receives 3x marginally below the acceptance threshold and 1x reject, not good enough. The reviews are mostly negative: The results shown do not clearly show a significant improvement compared to other methods. The authors use 3 datasets in which two of them are much smaller dataset (Mouse, Pranav) than the other one (FLIC). This raises the question of generalizability of this method to larger datasets. Technical contributions are not clear.